# Investigating and Modelling an Engineered Millifluidic In Vitro Oocyte Maturation System Reproducing the Physiological Ovary Environment in the Sheep Model

**DOI:** 10.3390/cells11223611

**Published:** 2022-11-15

**Authors:** Antonella Mastrorocco, Ludovica Cacopardo, Letizia Temerario, Nicola Antonio Martino, Federico Tridente, Annalisa Rizzo, Giovanni Michele Lacalandra, Domenico Robbe, Augusto Carluccio, Maria Elena Dell’Aquila

**Affiliations:** 1Department of Biosciences, Biotechnologies & Environment, University of Bari Aldo Moro, Via Edoardo Orabona, 70125 Bari, Italy; 2Research Centre E. Piaggio, University of Pisa, Largo Lucio Lazzarino 1, 56122 Pisa, Italy; 3Department of Veterinary Medicine, University of Bari Aldo Moro, Str. Prov. Casamassima Km. 3, 70010 Valenzano, Italy; 4Faculty of Veterinary Medicine, University of Teramo, Loc. Piano d’Accio, 64100 Teramo, Italy

**Keywords:** millifluidic culture, oocyte, in vitro maturation, mitochondria, intracellular reactive oxygen species, computational models, oxygen diffusion, ovary environment

## Abstract

In conventional assisted reproductive technologies (ARTs), oocytes are in vitro cultured in static conditions. Instead, dynamic systems could better mimic the physiological in vivo environment. In this study, a millifluidic in vitro oocyte maturation (mIVM) system, in a transparent bioreactor integrated with 3D printed supports, was investigated and modeled thanks to computational fluid dynamic (CFD) and oxygen convection-reaction-diffusion (CRD) models. Cumulus-oocyte complexes (COCs) from slaughtered lambs were cultured for 24 h under static (controls) or dynamic IVM in absence (native) or presence of 3D-printed devices with different shapes and assembly modes, with/without alginate filling. Nuclear chromatin configuration, mitochondria distribution patterns, and activity of in vitro matured oocytes were assessed. The native dynamic mIVM significantly reduced the maturation rate compared to the static group (*p <* 0.001) and metaphase II (MII) oocytes showed impaired mitochondria distribution (*p <* 0.05) and activity (*p <* 0.001). When COCs were included in a combination of concave+ring support, particularly with alginate filling, oocyte maturation and mitochondria pattern were preserved, and bioenergetic/oxidative status was improved (*p <* 0.05) compared to controls. Results were supported by computational models demonstrating that, in mIVM in biocompatible inserts, COCs were protected from shear stresses while ensuring physiological oxygen diffusion replicating the one occurring in vivo from capillaries.

## 1. Introduction

In vitro maturation (IVM) of oocytes is an assisted reproductive technology (ART), and an alternative to controlled ovarian hyperstimulation (COH). In COH, patients undergo pituitary downregulation followed by exogenous gonadotropin administration and retrieval of in vivo matured oocytes [1]. However, in IVM, meiotically immature oocytes are retrieved from unstimulated ovaries and cultured in vitro prior to fertilization under the influence of different additives, including but not limited to gonadotrophins. IVM has great potential to become the technique of choice in human and animal reproductive medicine, even though, despite its long history, it needs technical improvements.

In animal ARTs, it is the predominant method of maturation of oocytes used for in vitro production of embryos for commercial and research purposes [2]. The first report of oocyte isolation from ovarian follicles and IVM was published by Pincus and Enzmann (1935) in rabbit oocytes [3] and currently is widely used for artificial breeding, cloning, and transgenic animal production [4]. However, despite the fruitfulness of these technologies, the developmental rates of in vitro produced animal embryos derived from IVM oocytes are still suboptimal and species-specific [5].

The potential applications of IVM in human ARTs were also recognized by Edwards in his landmark 1965 publications [6,7]. The first live human birth following collection of immature oocytes from non-stimulated cycles and their maturation in vitro was recorded in 1991 [8]. Since then, more than 5000 live human births have occurred as a result of oocyte IVM [9]. In human ARTs, IVM is currently used in specific circumstances in which it is proposed in order to avoid side effects of gonadotropin administration, such as in patients at risk of ovarian hyperstimulation syndrome [10,11], or those requiring rapid fertility preservation strategies before undergoing chemo or radio therapy for estrogen-sensitive cancer [12,13,14]. Moreover, IVM allows for the minimization of COH adverse side effects in patients with polycystic ovarian syndrome and reduces the gonadotropin use and costs associated with drug use and patient monitoring [15]. IVM indications have been also expanded to rare conditions such as resistant ovary syndrome or repeated deficient oocyte maturation, where a COH failed to result in mature oocytes [16].

Although perspectives are promising, the clinical outcomes of IVM oocytes are lower compared with those of oocytes matured in vivo [17,18,19,20,21]. To improve IVM protocols, in vitro culture conditions should be optimized to better support nuclear and cytoplasmic modifications occurring physiologically as a result of in vivo ovulatory stimuli [22,23,24,25,26]. Cumulus-oocyte complex (COC) development and maturation are regulated by different factors, such as those arriving via the afferent vasculature, i.e., FSH, LH, those establishing interactions between granulosa-cumulus cells and the oocyte, and a variety of growth factors [27,28,29,30,31,32]. Interesting outcomes may arise from innovative strategies of dynamic cell culture devices purposely engineered to provide the cells with controlled media flow/movement by simulating continuous removal of harmful products and replacement of substrates, physical stimulation, and activation of signaling pathways [33]. As can be seen from current literature, considerable interest in the use of dynamic microfluidic systems has recently emerged. In vitro studies on evaluation and handling of reproductive cells and tissue, such as those of maternal–placental–embryonic axis [34], oocytes [35,36,37], sperm cells [37,38,39], human placental trophoblast [40,41], embryos [35,36,37], and embryo culture [42,43,44,45,46], have been reported. To the best of our knowledge, only two studies have been carried out to date on dynamic microfluidic IVM culture, both in immature oocytes retrieved from superovulated mice [47,48]. In these studies, this method reduced lipid peroxidation levels in oocyte culture media and apoptosis rate in matured oocytes [47] and increased oocyte maturation and blastocyst formation rate [48]. In farm animals, the only microfluidic IVM study published to date involved the use of a microfluidic but not dynamic device to trap and culture bovine COCs [49]. In large animal models, as well as in humans, due to the large ovary, follicle, COC size, and follicular fluid volume at the preovulatory stage [50], studies on millifluidic devices could be more appropriate to replicate the physiological environment. Indeed, the COC grows and matures in the follicular antrum, an environment that develops and become progressively larger, reaching dimensions measured in centimeters in diameter in the preovulatory maturation phase. During its development, although not exposed directly, but through the mediation of theca and granulosa cells, the COC receives nutrients and oxygen supply by continuous diffusion from the microcapillary network that surrounds the follicle. Ovarian blood flow rates range from milliliters/minute (at ovarian arterial level) to microliters/minute (at follicular capillary beds of antral follicles) [51,52,53,54]. In a previous study, we found that millifluidic IVM (mIVM) improved oocyte mitochondrial membrane potential, intracellular ROS levels, and mitochondria/ROS colocalization, markers of healthy oocyte cytoplasmic maturation. In that study, a previously developed 3D mIVM system with COC-including bioprinted alginate microbeads was used [55] and inserted into the millifluidic device [56].

The aims of the present study were to analyze the effects of millifluidics on IVM in a large animal model, such as the sheep. Culture was performed in a transparent bioreactor under controlled flow rates by comparing three COC culture conditions, i.e., (1) native COCs with no artificial coating, (2) COCs included in 3D-printed biocompatible modular supports, and (3) COCs embedded in an alginate-filled supports. The maturation rate and bioenergetic/oxidative parameters of matured oocytes were assessed and obtained data were validated through computational models of fluid dynamics and oxygen diffusion.

## 2. Materials and Methods

### 2.1. Chemicals

All chemicals for in vitro cultures and analyses were purchased from Sigma-Aldrich (Milan, Italy), unless otherwise indicated.

### 2.2. Collection of Ovaries and COC Retrieval

Ovaries from prepubertal lambs (less than 6 months of age) were recovered at local slaughterhouses, from the animals subjected to routine veterinary inspection in accordance with the specific health requirements stated in Council Directive 89/556/ECC and subsequent modifications. Ovaries were transported to the laboratory at room temperature within 4 h after collecting. For COC retrieval, ovaries underwent the slicing procedure [57]. Follicular contents were released in sterile Petri dishes containing phosphate buffered saline (PBS) and observed under a Nikon SMZ18 stereomicroscope equipped with a transparent heating stage set up at 37 °C (Okolab S.r.l., Napoli, Italy). Only COCs with at least three intact cumulus cell layers and homogenous cytoplasm were selected for culture [56].

### 2.3. In Vitro Maturation (IVM)

IVM medium was prepared based on TCM-199 medium with Earle’s salts. It was buffered with 5.87 mmol/L HEPES and 33.09 mmol/L sodium bicarbonate, and supplemented with 0.1 g/L L-glutamine, 2.27 mmol/L sodium pyruvate, calcium lactate pentahydrate (1.62 mmol/L Ca^2+^, 3.9 mmol/L Lactate), 50 μg/mL gentamicin, 20% (*v*/*v*) fetal calf serum (FCS), 10 μg/mL of porcine follicle stimulating hormone (FSH), luteinizing hormone (LH; Pluset^®^, Calier, Balcellona, Spain) [58], and 1μg/mL 17β estradiol [57]. IVM medium was pre-equilibrated for 1 h under 5% CO_2_ in air at 38.5 °C, then loaded (400 µL/well) in a 4-well dish (Nunc Intermed, Roskilde, Denmark) and covered with pre-equilibrated lightweight paraffin oil. In each experiment, 20–25 COCs/well were added to a 4-well dish and cultured under conventional static IVM as control (CTRL) or loaded in a chamber of a commercial Live Box 1 (LB1) bioreactor (IVTech S.r.l.—Massarosa, Italy) and used to test different mIVM systems, as described in Table 1. In each experiment, in vitro culture was performed by placing the bioreactor, together with the control 4-well plate, in the incubator for 24 h at 38.5 °C under 5% CO_2_ in air.

### 2.4. Native Millifluidic IVM (Native mIVM)

In the first experiment, mIVM was performed by inserting COCs as they are (in the further text “native COCs”), on the bottom of the culture chamber of a commercial LB1 bioreactor. This culture system has been called “native mIVM” in the further text (Table 1). The LB1, consisting of a polydimethylsiloxane (PDMS) chamber with a transparent top and bottom, was designed to reproduce the typical volume of the single well of a 24-well plate (2 mL) with a flow inlet and an outlet for the perfusion of culture media, was assembled as described by Mastrorocco et al. (2021) [56]. Briefly, COCs were placed inside the cell chamber after adding 300 μL of IVM medium. Then, the chamber was hermetically sealed and filled with an additional 1700 μL of IVM medium through the inlet tube. The bioreactor was kept for two hours in the incubator to allow the COCs to stabilize at the bottom of the chamber before flow activation. Two hours after COC loading, the culture chamber was connected to a millifluidic circuit composed of a mixing chamber, a reservoir of 18 mL with IVM medium, and a peristaltic pump which permits the circulation of the culture medium. A silicone tube connected with a 0.22 µm filter, in the mixing chamber, allowed medium oxygenation. The flow rate was temporally set at 450 μL/min for the time to quickly fill the circuit and then regulated to 50 or 100 μL/min, according to the experimental design. Flow rates were chosen based on previous studies in the sheep which reported that ovarian blood flow gradually decreases from the order of milliliters/min in the ovarian artery [51,52], to microliters/min in the capillary network around the developing follicles, varying in relation to various aspects such as the anatomical district and the ovarian cycle stage [53,54].

### 2.5. mIVM with 3D Printed Biocompatible Supports

In the second experiment, mIVM culture was performed by inserting COCs within biocompatible supports placed at the bottom of the bioreactor chamber in order to recreate a protecting structure around them, similar to that of an ovarian follicle or part of it (Table 1). The supports, consisting of a ring (inner/outer radius = 5.5/7 mm) and a plain or a concave disk (outer radius 7 mm, maximum depth = 1.5 mm), were designed with Fusion 360 and fabricated at the Research Center ‘E. Piaggio’, University of Pisa, using a stereolithographic 3D printer (Form2, FormLabs) loaded with a biocompatible photo-polymeric material (Dental SG resin, Formlabs) [59]. The configurations tested were: (i) ring; (ii) concave + ring; (iii) concave + ring + plain (namely ring, concave + ring and “box” in the further text; Table 1; Figure 1). In all conditions, COCs were uploaded in the chamber after inserting the supports and adding the first 300 µL of IVM medium. Next, the chamber was then completely filled, closed, and connected to the pump as described above. The flow rate was set at 50 μL/min.

### 2.6. mIVM in Biocompatible Supports Filled with Alginate Gel

In the third experiment, a concave support and a ring were inserted at the bottom of the bioreactor chamber and filled with alginate gel with the aim to reproduce 3D conditions for COC culture. Inside the support, 150 µL of 1% *w*/*v* sodium alginate pre-equilibrated in IVM medium was released [55] and, within it and appropriately spaced, 20–25 COCs were uploaded. This culture condition was named “alginate-filled support” (Table 1). A 100 mM calcium chloride (CaCl_2_) solution was then sprayed onto the surface of the gel at the top edge of the concave support [60]. After this step, CaCl_2_ was added in droplets to allow complete alginate jellification and COC trapping. After 5 min, CaCl_2_ was removed. The bioreactor was filled with IVM medium, sealed and kept for two hours in the incubator. After that, it was connected to the reservoir of IVM medium, and a pump set to a flow rate of 50 µL/min started the millifluidic culture as described above. In each replicate, 20–25 COCs were cultured. After 24 h of IVM culture, sodium alginate was removed by calcium chelation with 2% *w*/*v* sodium citrate in IVM medium for 5 min at 38.5 °C under 5% CO_2_ in air.

### 2.7. Assessment of Cumulus Expansion and Oocyte Denuding

After all three types of experiments, COCs were recovered and cumulus expansion was checked. COCs showing cumuli with continuous edges, consisting of cells in close contact each other, were classified as compact, whereas cumuli showing discontinuous edges following cell detachment and production of a viscous extracellular matrix were classified as expanded. The percentage of expanded COCs was noticed, as it represents a response of immature oocytes to the presence of gonadotropins in the medium even if it does not fully ensure that the maturation is achieved. COCs underwent cumulus cell removal by incubation in TCM-199 with 20% FCS containing 80 IU hyaluronidase/mL and aspiration in and out of finely drawn glass pipettes. Denuded oocytes were evaluated for meiotic stage and matured ones were used to assess bioenergetic/oxidative status.

### 2.8. Oocyte Mitochondria and ROS Staining

Oocytes were washed three times in PBS with 3% BSA and incubated for 30 min in the same medium containing 280 nmol/L MitoTracker Orange CMTM Ros (Thermo Fisher Scientific, Waltham, MA, USA) at 38.5 °C under 5% CO_2_ in air. After incubation with MitoTracker, oocytes were washed in PBS with 0.3% BSA and incubated for 15 min, at 38.5 °C under 5% CO_2_, in air in the same medium containing 10 µmol/L 2′,7′-dichlorodihydrofluorescein diacetate (H_2_DCF-DA) to detect the dichlorofluorescein (DCF) and localize intracellular sources of ROS [61]. After incubation, oocytes were washed in PBS without BSA and fixed overnight at 4 °C in 4% paraformaldehyde (PFA) solution in PBS [62]. Particular attention was applied to avoid sample exposure to the light during staining and fixing procedures and to reduce photobleaching.

### 2.9. Oocyte Nuclear Chromatin Evaluation

To evaluate oocyte nuclear chromatin, after the fixation in 4% PFA in PBS, oocytes were stained with 2.5 μg/mL Hoechst 33,258 in 3:1 (*v*/*v*) glycerol/PBS mounted on microscope slides with coverslips, sealed with nail polish, and kept at 4 °C in the dark until observation. Slides were examined under the epifluorescence microscope (Nikon Eclipse 600; Nikon Instruments, Firenze; ×400 magnification) equipped with a B-2A (346 nm excitation/460 nm emission) filter. Oocytes were evaluated in relation to their meiotic stage, and classified as germinal vesicle (GV), metaphase to telophase I (MI to TI) and MII with the first polar body extruded [63]. Oocytes showing either multipolar meiotic spindle, irregular chromatin clumps, or absence of chromatin were considered as abnormal [55].

### 2.10. Assessment of Mitochondrial Distribution Pattern and Intracellular ROS Localization

Oocytes at the MII stage were observed at ×600 magnification in oil immersion with a Nikon C1/TE2000-U laser scanning confocal microscope (Nikon Instruments, Firenze). A 543 nm helium/neon laser and the G-2A filter were used to detect the MitoTracker Orange CMTM Ros (551 nm excitation and 576 nm emission). A 488 nm argon ion laser and the B-2A filter were used to detect DCF (495 nm excitation and 519 nm emission). Scanning was conducted with 25 optical sections from the top to the bottom of the oocytes, with a step size of 0.45 μm to allow 3D distribution analysis. The mitochondrial distribution pattern was evaluated on the basis of previous studies: (1) finely granular, with small mitochondria aggregates spread throughout the cytoplasm, typical of immature oocytes; (2) perinuclear and subplasmalemmal (P/S) distribution of mitochondria forming large granules, which is indicator of cytoplasmic maturity; (3) abnormal, with irregular distribution of mitochondria [62]. Concerning intracellular ROS localization, oocytes with intracellular ROS distributed throughout the cytoplasm, together with areas/sites of mitochondria/ROS overlapping, were considered healthy.

### 2.11. Quantification of Mitochondrial Activity, Intracellular ROS Levels, and Mitochondria-ROS Colocalization

In each individual oocyte, MitoTracker and DCF fluorescence intensities were measured at the equatorial plane and at the excitation/emission, as described above by use of EZ-C1 Gold Version 3.70 image analysis software platform for Nikon C1 confocal microscope. A circular area was drawn in order to measure only the region including cell cytoplasm. The fluorescence intensity within the programmed scan area was recorded and plotted against the conventional pixel unit scale (0–255). Mitochondrial activity and intracellular ROS levels were recorded as MitoTracker Orange CMTM Ros and DFC fluorescence intensity in arbitrary densitometric units (ADU). Parameters related to fluorescence intensity, such as laser energy, signal detection (gain), and pinhole size, were maintained at constant values for all measurements. The degree of mitochondria-ROS colocalization, reported as a biomarker of healthy oocytes [62,63] was quantified by the overlap coefficient between MitoTraker Orange CMTM Ros and DCF fluorescence intensity signals.

### 2.12. Computational Models of Millifluidic IVM Systems

To evaluate the effects of flow on COC in vitro cultures, computational models of tested culture systems were developed in collaboration with the Research Center ‘E. Piaggio’ of the University of Pisa. The models were implemented in COMSOL Multiphysics (Stockholm, Sweden), coupling incompressible Navier–Stokes and CRD equations. The bioreactor was modeled as a cylindrical chamber with PDMS lateral walls (oxygen permeability = 3.78 × 10^−11^ mol m m^−2^ s^−1^ mmHg) [64,65], while the static well was implemented as a cylindrical geometry with a media layer of 2.1 mm covered by an oil layer of 2.1 mm. The support geometries were imported from the Fusion 360 files and placed at the bottom of the bioreactor reproducing the different configurations investigated. The fluid viscosity of the culture medium was assumed equal to that of water at 37 °C (i.e., 0.6913 mPa × s), while the oxygen diffusion coefficient was assumed equal to 3 × 10^−9^ in the media and 2.5 x 10^−9^ m^2^/s in the alginate gels [66] and to 2 × 10^−9^ m^2^/s in the oil layer [67]. Finally, COCs were modeled as spheres with 0.3 mm radius (number = 21) and a consumption rate (OCR= 4 pmol/min) estimated from data reported in the literature for bovine COCs [68]. The initial oxygen concentration and the oxygen partial pressure were set at 0.21 mol/m^3^ and 159 mmHg, respectively.

### 2.13. Statistical Analysis

The proportions of oocytes showing different chromatin configurations and mitochondria distribution patterns were compared between groups by the Chi-square test. Mitochondria and ROS quantification analysis was conducted on oocytes at MII stage. Data (mean ± standard deviation [SD]) were compared by one-way ANOVA followed by Dunnett’s Multiple Comparison Test. Differences with *p <* 0.05 were statistically significant.

## 3. Results

### 3.1. Native mIVM Did Not Sustain Oocyte Nuclear and Cytoplasmic Maturation

In experiment 1, in three independent runs, 229 COCs were cultured and 222 of them were evaluated. Under both flow rates of 50 and 100 µL/min, after 24 h mIVM, COCs were found at the edges of the bioreactor chamber. The cumulus expansion rate was 25% compared to 100% of COCs cultured in control static IVM. In mIVM, the percentage of oocytes which reached the MII stage was significantly lower than in controls (*p <* 0.001, Table 2). Conversely, the percentage of oocytes that remained at the MI-TI stages was significantly increased either at 50 µL/min (*p <* 0.01, Table 2) or at 100 µL/min (*p <* 0.001, Table 2). Moreover, a statistically significant increase of the percentage of oocytes showing abnormal chromatin configurations, such as multipolar meiotic spindle or irregular chromatin clumps, was observed (*p <* 0.05, Table 2).

In most of the control MII oocytes, the mitochondria distribution was P/S, which is a biomarker of cytoplasmic maturity (Table 3). Instead, after mIVM, either under 50 or 100 µL/min flow rate, significantly higher proportions of MII oocytes with finely granular mitochondria distribution were found (*p <* 0.05 and *p <* 0.01, respectively; Table 3 and Figure 2) with corresponding reduction of the percentage of oocytes with healthy P/S pattern (*p <* 0.05 and *p <* 0.001, respectively; Table 3 and Figure 2). At both tested conditions, oocyte mitochondrial membrane potential, ROS levels, and mitochondria/ROS colocalization were significantly reduced compared with controls (*p <* 0.001; Figure 2).

### 3.2. mIVM with Open Biocompatible Supports Sustain Oocyte Nuclear and Cytoplasmic Maturation

Based on the results of experiment 1, in Experiment 2, the effects of including the COCs in biocompatible supports placed inside the culture chamber, were tested. The oocytes were cultured in six replicates. In each replicate, at least one or two test conditions were examined and compared to the controls. A total of 372 COCs were cultured and 348 COCs were analyzed. By using the ring support, after mIVM, cumulus expansion was 50% compared with 100% of control COCs. The percentage of oocytes reaching the MII stage was lower compared to controls (*p <* 0.05; Table 4). When the concave support and the ring support were tested, cumulus expansion reached 80% and the percentage of matured oocytes returned to be comparable to that of static controls. Instead, in the box support, cumulus expansion was reduced (20%) and oocyte nuclear maturation was inhibited (*p <* 0.001, Table 4). Figure 3 shows COCs cultured under native mIVM (A), with the ring (B) and the concave + ring (C) supports.

Both the mature oocytes obtained by mIVM culture in the presence of ring and concave + ring supports showed healthy P/S type mitochondrial distribution pattern comparable to the controls (Table 5). On the contrary, after mIVM in the box support, the percentage of matured oocytes with P/S mitochondrial distribution pattern was significantly reduced (*p <* 0.05, Table 5), with a significantly higher percentage of matured oocytes showing abnormal, particularly large mitochondria granulations spread all over the cytoplasm or located in specific cytoplasmic domains (*p <* 0.001, Table 5 and Figure 4). In oocytes matured with the ring support, mitochondrial activity, ROS levels, and mitochondria/ROS colocalization were significantly reduced compared to controls (*p <* 0.001; Figure 4). In MII oocytes derived from mIVM with concave + ring, no differences were noticed for mitochondrial activity and ROS levels compared with controls. However, as both values were higher than controls, a statistically significant increase in mitochondria/ROS colocalization was found (*p <* 0.001; Figure 4). For oocytes matured in the box support, quantification parameters did not differ with those of controls.

### 3.3. D-mIVM in Alginate Layer Sustain Oocyte Nuclear and Cytoplasmic Maturation

In Experiment 3, the effects of mIVM with COCs included in supports filled with an alginate layer were analyzed. A total of 150 oocytes were processed in three independent runs and 146 of them were analyzed. After 24 h of IVM in alginate-filled supports, cumulus expansion ranged between 80 and 100%, and the percentage of MII oocytes was comparable to that of controls (Table 6). Additionally, no differences in mitochondria distribution pattern were found compared to controls (Table 7). These MII oocytes showed increased mitochondrial activity (*p <* 0.05), ROS levels (*p <* 0.001; Figure 5), and mitochondria/ROS colocalization (*p <* 0.05; Figure 5). Figure 6 shows representative photomicrographs of oocytes cultured in the analyzed condition. It can be seen that oocytes matured in alginate-filled supports showed P/S mitochondria distribution pattern and increased mitochondria- and ROS-related fluorescence intensities.

### 3.4. Shear Stress and Oxygen Concentration Modulation in the Different IVM Configurations

In the native mIVM, maximum shear stress values on the COCs were higher at 100 μL/min (6.18 × 10^−6^ Pa) than 50 μL/min (3.25 × 10^−6^ Pa). Average oxygen concentrations were higher (0.196 mol/m^3^, for both 50 and 100 μL/min) compared to the static well controls (0.188 mol/m^3^).

In the ring and concave + ring configurations, shear stress values were comparable with the bioreactor set at 50 μL/min in native conditions (3.14 × 10^−6^ and 3.28 × 10^−6^ Pa) and average oxygen concentrations were not significantly increased with respect to static conditions (0.184 and 0.182 mol/m^3^, respectively). Differently, in the box configuration, despite protection of COCs from flow and shear stress, oxygen concentration was lower (0.146 mol/m^3^).

Finally, in the alginate-filled support, due to the presence of alginate, direct flow rate was absent and so the shear stress and average oxygen levels resulted as being equal to 0.182 mol/m^3^.

Figure 7 shows the CFD and oxygen CRD models and relative data for all the examined conditions.

## 4. Discussion

The first step of this study was to investigate whether the native mIVM, without any kind of protective or 3D-inclusive support, could be advantageous for COC maturation. This hypothesis was formulated on the basis on previous studies in other cell systems, which showed that millifluidic cell cultures improve nutrient and oxygen supply [65,69,70,71]. To the best of our knowledge, this is the first study on native mIVM in sheep, a large animal model. In a previous study from our unit, we found that mIVM, not even increasing the maturation rate, boosted oocyte bioenergetic/oxidative status compared with the static conditions. However, in that study, a 3D-IVM system was used by constructing COC-including alginate microbeads via bioprinting technologies [56]. Differently, in this study, we wanted to test direct effects of mIVM itself on the COC. To our knowledge, this is the first study analyzing the effect of dynamic mIVM on immature oocytes of farm animals not submitted to superovulation treatments but derived from the spontaneous ovarian cycle, which are suitable models for fertility rescue or preservation programs in both humans and in animals. Previous studies were performed in oocytes from superovulated mice [47,48] or in bovine oocyte by using a microfluidic but not dynamic system [49]. The results of the first experiment clearly indicated that exposure of native COCs to the tested flow conditions did not support oocyte nuclear and cytoplasmic maturation, as most of the oocytes did not reach the MII stage, and a few matured oocytes showed compromised bioenergetic/oxidative status.

In order to evaluate the reasons for such significant oocyte damage, computational models of the tested culture systems were developed. As shown by the CFD models, the shear stress on COCs was found to increase correspondingly to the flow rate. Moreover, according to the CRD computation, higher average oxygen concentrations were found with respect to static well controls. Thus, these conditions may have negatively affected COC maturation, and may be responsible for observed inhibition of oocyte meiotic maturation and mitochondrial unbundling and reduced activity. Indeed, in vivo, COCs are not directly exposed to blood flow and nutrients, and oxygen reaches them by diffusion from capillaries. It is worth mentioning that ovaries have a complex blood supply system required to support ovarian functions. For example, in the sheep model as well as in humans, growing follicles are found in the corticomedullary border, a region particularly well-supplied with blood vessels [72]. Follicle microvasculature never passes beyond the basal lamina before ovulation when it invades into the granulosa layer [73,74]. The ovarian vasculature is involved in oxygen tension regulation as oxygen released from capillaries first reaches the thecal cells and then lower amounts diffuse through the basement membrane to reach the multiple layers of granulosa cells and the COC. The oxygen concentration at the follicle surface in developing human follicles is predicted with a range of 0.121-0.131 mol/m^3^ [50,75]. Due to the lack of direct blood supply, as the diameter of ovarian follicles greatly increases during follicle maturation, the diffusion distance for gasses inside the follicle also increases, and this led to a continuous decrease of oxygen concentration in the follicular fluid during follicular maturation, reaching the lowest levels in preovulatory follicles [50,76,77]. These data are in line with compromised COC maturation found under flow rate and oxygen concentrations higher than those of physiological conditions.

Based on the results of experiment 1 and information of in vivo condition reported in the literature, in experiment 2, we evaluated whether the insertion of the COCs in biocompatible supports, placed inside the culture chamber, could protect the COCs from the shear stress and from possible side effects of high oxygen levels, maintaining the advantages of millifluidics (i.e., continuous arrival of new nutrients, the removal of metabolism waste). By inserting the ring support, less significant cell damage was observed. In fact, the percentages of nuclear maturation, although reduced compared to the controls, were almost high as absolute value (53%) and significantly higher than that of IVM culture in the absence of supports at the same flow rate (53% in Table 4 vs 27% in Table 2 at 50 μL/min; *p <* 0.05). Furthermore, the good P/S mitochondrial pattern was preserved in a percentage comparable to the controls, although the quantification data indicated a significant trend towards loss of viability. More evident improvement was obtained by inserting two supports, the concave one and the ring one, as the nuclear maturation and P/S mitochondrial pattern were reached at percentages comparable to the controls. Additionally, quantification data of mitochondria activity and ROS levels tended to increase, indicating cellular activity and viability, even if they did not attain statistical significance, which was observed for colocalization. Conversely, the box configuration significantly inhibited oocyte nuclear maturation and mitochondria distribution, although quantification parameters were not affected. Computational models revealed that, in the support configurations (ring and concave + ring), although shear stress values were comparable with those of the native mIVM culture at 50 μL/min without supports, oxygen concentration did not significantly increase. Moreover, in both configurations, at the end of the culture time COCs were found to be much closer together. These two conditions, lower oxygen levels and increased proximity between COCs, may have had a positive impact on COC culture conditions (e.g., by promoting the exchange of signaling molecules). In addition, the concave support may also have had a positive effect replicating in vivo like curvatures as in previous studies [78]. Differently, in the box configuration, despite COCs being protected from flow, oxygen concentrations were lower. Indeed, the box configuration had a low oxygen permeability which may have inhibited oxygen diffusion towards the COCs, thus affecting their major cellular activities. Moreover, the box configuration may negatively affect waste removal and nutrients supply. In agreement with these observations, in vitro studies demonstrated that prevailing low oxygen levels can impact on bovine granulosa cell, inducing their early luteinization with down-regulation of FSH signaling, cell proliferation and steroidogenesis beside up-regulation of angiogenesis, glucose metabolism, and inflammatory processes [79].

Finally, the alginate-filled configuration seems to have been the best analyzed condition, since the oocytes that matured in this condition showed an improvement in their oxidative/bioenergetic status, even at a maturation rate comparable with controls. The higher COC proximity, concave support curvature, 3D environment provided by the gel [80], and the absence of direct flow and basal oxygen levels may have recreated an environment similar to that of the ovarian follicle [80].

In conclusion, the obtained results demonstrate that, in the sheep model, mIVM with biocompatible supports protects the COCs from direct medium flow and elevated oxygen concentrations while keeping optimal nutrient supply and waste removal. This result is a prelude to an improvement of the developmental competence of oocytes matured under mIVM. This culture method could be applied to oocytes in suboptimal conditions, such as those recovered from ovine breeds under risk of genetic erosion, from other animal endangered species, from aged donors, or to oocytes matured after cryopreservation.

## Figures and Tables

**Figure 1 cells-11-03611-f001:**
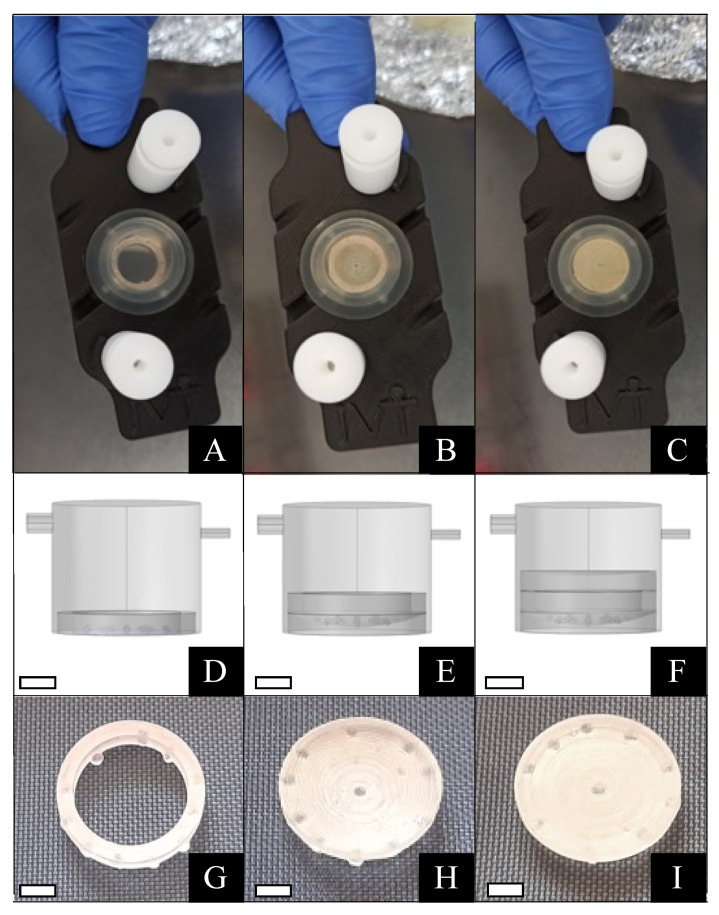
Set up of mIVM with biocompatible supports. In the higher section, an upper view of the bioreactor chamber with the ring (**A**), the concave + ring (**B**), and the completely assembled box support (**C**) at the bottom of the chamber. In the middle section, a schematic drawing of the bioreactor chamber set up with the ring support (**D**), the concave support with the ring on the top of it (**E**), and the assembled box (**F**) placed at the bottom of the chamber. In the lower section, close looks at the ring (**G**), concave (**H**), and plain disk (**I**) supports. Scale bars representing 3.5 mm for D–I pictures.

**Figure 2 cells-11-03611-f002:**
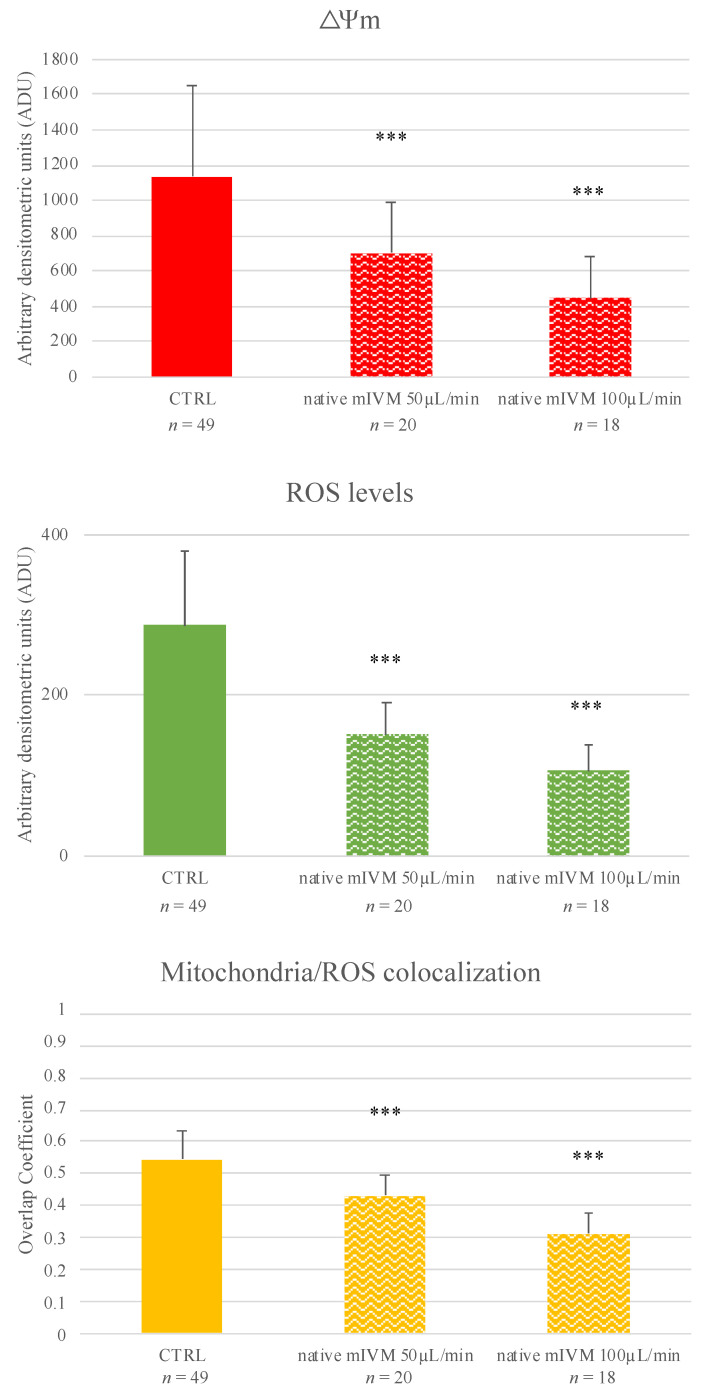
Effects of native mIVM performed at 50 or 100 μL/min on mitochondria activity, intracellular ROS levels, and mitochondria/ROS colocalization of sheep MII oocytes. Numbers of analyzed matured oocytes are indicated at the bottom of each graph bar. One-way ANOVA, Dunnett’s Multiple Comparison Test: comparisons mIVM versus control *** *p <* 0.001.

**Figure 3 cells-11-03611-f003:**
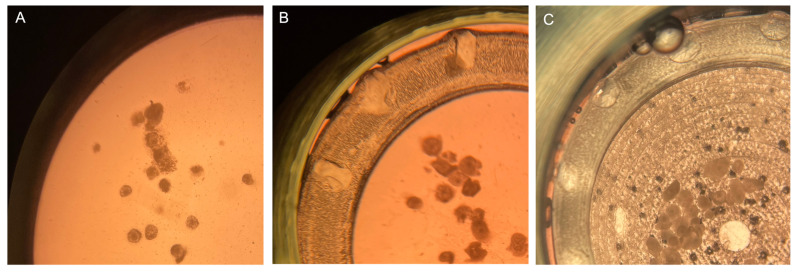
Representative photomicrographs of sheep COCs after culture under: (**A**) native mIVM; (**B**) by using the ring support and (**C**) in the presence of the biocompatible supports concave + ring.

**Figure 4 cells-11-03611-f004:**
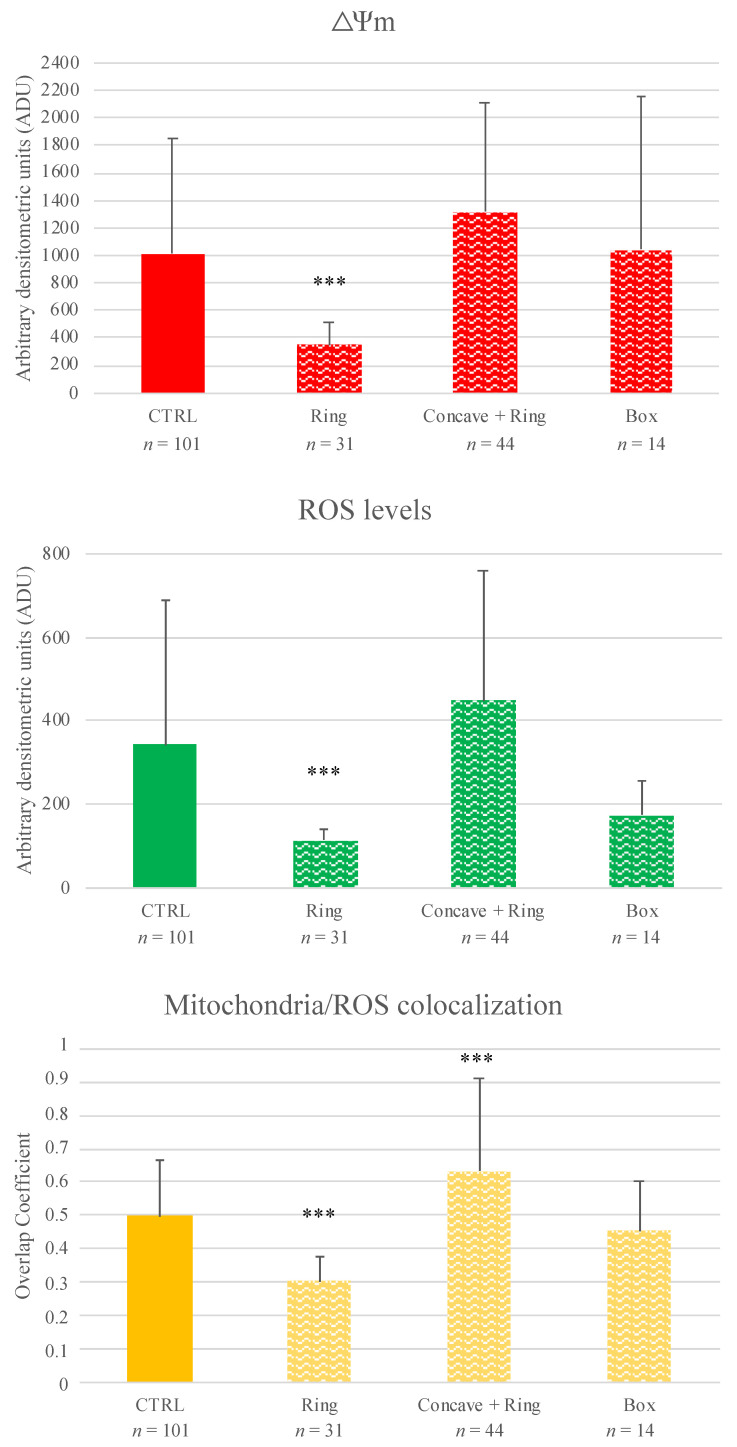
Effects of mIVM with biocompatible (ring; concave + ring; box) supports on mitochondria activity, intracellular ROS levels and mitochondria/ROS colocalization of sheep MII oocytes. Numbers of analyzed matured oocytes are indicated at the bottom of each graph bar. One-way ANOVA, Dunnett’s Multiple Comparison Test: comparisons mIVM versus control *** *p <* 0.001.

**Figure 5 cells-11-03611-f005:**
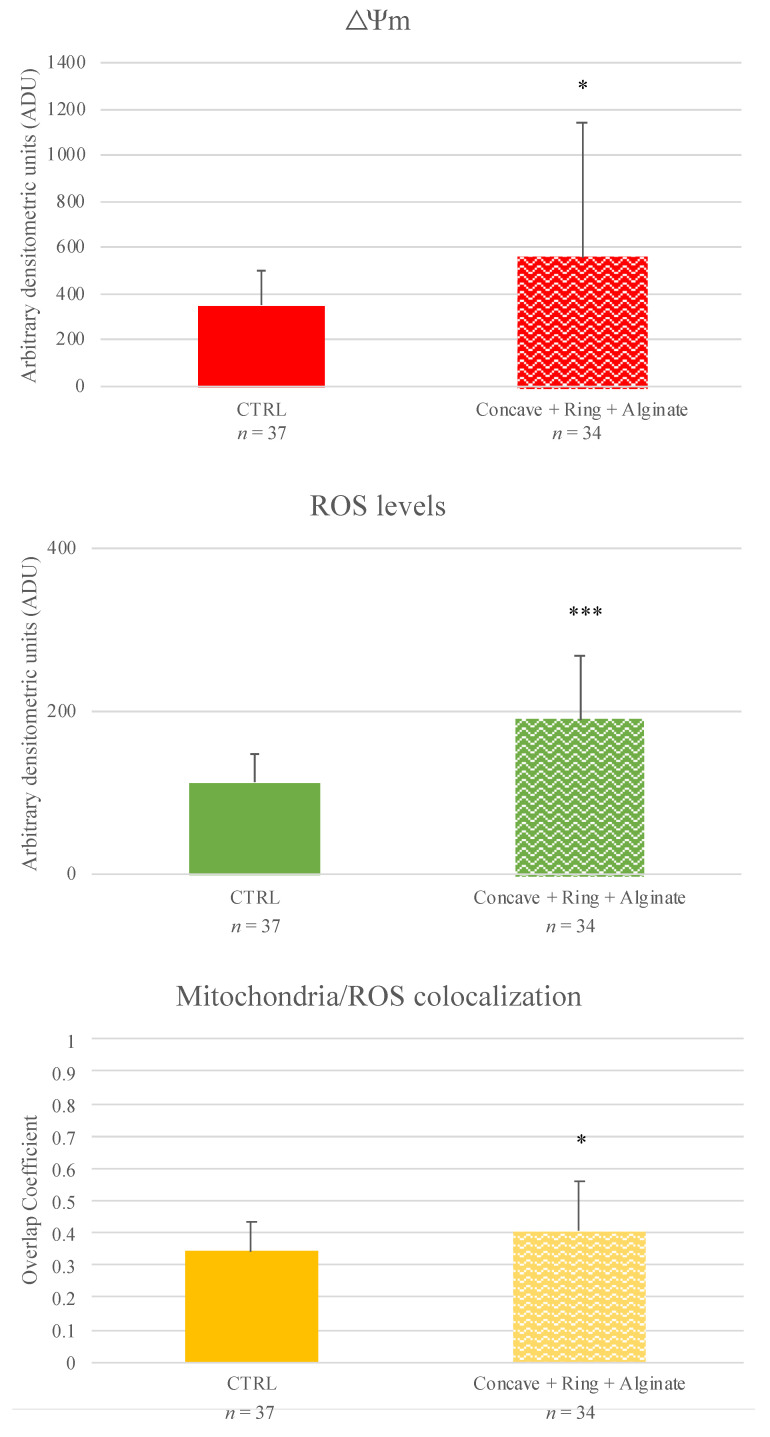
Effects of mIVM in alginate-filled supports on MII sheep oocyte mitochondria activity, intracellular ROS levels, and mitochondria/ROS colocalization. Numbers of analyzed matured oocytes are indicated at the bottom of each graph bar. Student’s *t*-test: * *p <* 0.05; *** *p <* 0.001.

**Figure 6 cells-11-03611-f006:**
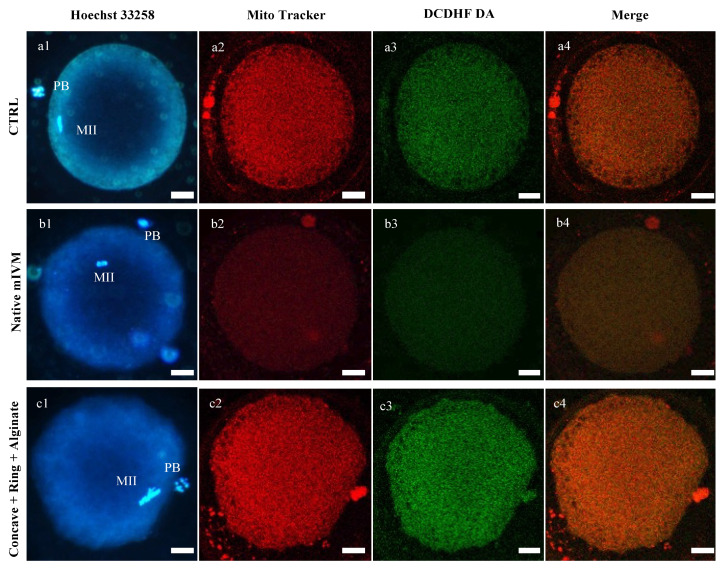
Effects of mIVM on sheep oocyte bioenergetic/oxidative status. Photomicrographs showing representative images of matured oocytes derived from static IVM (CTRL; (**a**)); native mIVM (**b**); in alginate-filled supports (**c**). Corresponding epifluorescence images showing nuclear chromatin configuration (column 1: Hoechst 33258) and confocal images showing perinuclear and subplasmalemmal (P/S; (**a**,**c**)) or finely granular (**b**) mitochondrial distribution pattern and activity (column 2: MitoTracker Orange CMTM Ros), intracellular ROS localization and levels (column 3: DCDHF DA), and mitochondria/ROS colocalization (column 4: Merge). Confocal images were taken at the oocyte equatorial plane. Scale bars represent 40 μm.

**Figure 7 cells-11-03611-f007:**
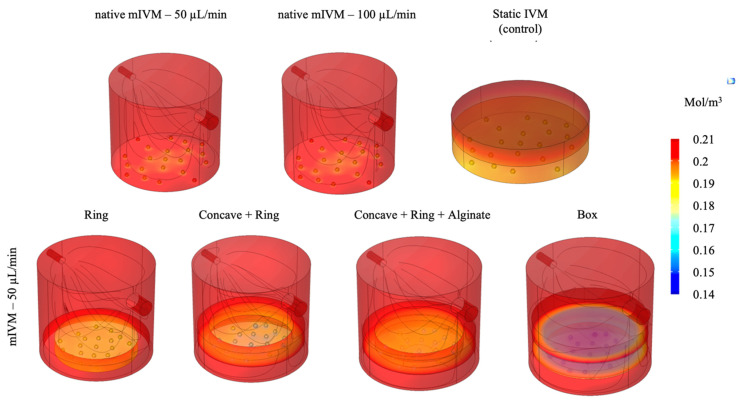
Computational fluid dynamic (CFD) and oxygen convection-reaction-diffusion (CRD) models of the examined conditions. Grey lines show the flow streamlines and the right bar indicates oxygen concentration in color scale from red (maximum concentration) to blue (minimum concentration).

**Table 1 cells-11-03611-t001:** Nomenclature of used millifluidic IVM (mIVM) culture systems.

Name	Description
Control	Static IVM in 4-well dish with no inserts.
Native	Dynamic mIVM in bioreactor without inserts.
Ring	Dynamic mIVM in bioreactor with and ring support placed at the bottom of the bioreactor chamber.
Concave + ring	Dynamic mIVM in bioreactor with concave support placed at the bottom of the bioreactor chamber and the ring support placed on its top.
Concave + ring + plain = Box	Dynamic mIVM in bioreactor with a box obtained by sequentially assembling the concave support at the bottom of the bioreactor chamber, the ring on its top and the plain support used as a lid.
Concave + ring + alginate = Alginate-filled supports	Dynamic mIVM in bioreactor with COCs embedded in alginate gel loaded in concave support and ring inserted at the bottom of the bioreactor chamber.

**Table 2 cells-11-03611-t002:** Effects of native mIVM on sheep oocyte nuclear maturation.

Flow Rate	n. of	n. of	Nuclear Chromatin Configurations n. (%)			
	Cultured COCs	Analyzed				
(µL/min)	(Runs)	COCs	GV	MI to TI	MII	Abnormal
0 (CTRL)	76 (3)	75	12	8 ^a^	49 ^a^	6 ^a^
			(16)	(11)	(65)	(8)
50	78 (3)	75	14	24 ^c^	20 ^d^	17 ^b^
			(19)	(32)	(27)	(23)
100	75 (3)	72	12	27 ^d^	18 ^d^	15 ^b^
			(17)	(38)	(25)	(21)

Table legend: GV: Germinal Vesicle; MI: Metaphase I; TI: Telophase I; MII: Metaphase II. Chi square test: ^a^,^b^: *p <* 0.05; ^a^,^c^: *p <* 0.01; ^a^,^d^: *p <* 0.001.

**Table 3 cells-11-03611-t003:** Effects of native mIVM on mitochondria distribution pattern of sheep MII oocytes.

Flow Rate	n. of	Mitochondria Distribution Pattern n. (%)		
	Analyzed			
(μL/min)	Oocytes			
	(Runs)	Finely Granular	Perinuclear/ Subcortical	Abnormal
0 (CTRL)	49 (3)	16 ^a^	31 ^a^	2 ^a^
		(33)	(63)	(4)
50	20 (3)	14 ^b^	5 ^b^	1
		(70)	(25)	(5)
100	18 (3)	15 ^c^	2 ^d^	1 ^b^
		(83)	(10)	(6)

Chi square test: comparisons CTRL vs mIVM: ^a^,^b^ = *p <* 0.05; ^a^,^c^ = *p <* 0.01; ^a^,^d^= *p <* 0.001.

**Table 4 cells-11-03611-t004:** Effects of mIVM with biocompatible supports on sheep oocyte nuclear maturation.

Support Type	n. of Analyzed COCs (Runs)	n. of Evaluated COCs	Nuclear Chromatin Configurations n. (%)			
			GV	MI to TI	MII	Abnormal
CTRL	150 (6)	144	20 ^a^	9 ^a^	101 ^a^	14
			(14)	(6)	(70)	(10)
Ring	70 (3)	59	6	12 ^c^	31 ^b^	10
			(10)	(20)	(53)	(17)
Concave + Ring	75 (3)	69	2 ^b^	19 ^d^	44	4
			(3)	(27)	(64)	(6)
Concave + Ring + Plain (Box)	77 (3)	76	33 ^d^	18 ^d^	14 ^d^	11
			(43)	(24)	(18)	(15)

Table legend: GV: Germinal Vesicle; MI: Metaphase I; TI: Telophase I; MII: Metaphase II. Chi square test: ^a^,^b^: *p* < 0.05; ^a^,^c^: *p* < 0.01; ^a^,^d^: *p* < 0.001.

**Table 5 cells-11-03611-t005:** Effects of mIVM with biocompatible supports on mitochondria distribution pattern of sheep MII oocytes.

Supports	n. of Analyzed Oocytes (Runs)	Mitochondria Distribution Pattern n. (%)		
		Finely Granular	P/S	Abnormal
CTRL	101 (6)	51	47 ^a^	3 ^a^
		(50)	(47)	(3)
Ring	31 (3)	20	11	0
		(65)	(35)	(0)
Concave + Ring	44 (3)	26	18	0
		(59)	(41)	(0)
Concave + Ring + Plain (Box)	14 (3)	8	1 ^b^	5 ^d^
		(57)	(7)	(36)

Chi square test: ^a^,^b^: *p <* 0.05; ^a^,^d^: *p <* 0.001.

**Table 6 cells-11-03611-t006:** Effects of mIVM in alginate-filled supports on sheep oocyte nuclear maturation.

Alginate	n. of Analyzed COCs (Runs)	n. of Evaluated COCs	Nuclear Chromatin Configurations n. (%)			
			GV	MI to TI	MII	Abnormal
CTRL	75 (3)	74	11	10	37	16 ^a^
			(15)	(13)	(50)	(22)
+	75 (3)	72	11	16	38	7 ^b^
			(15)	(22)	(53)	(10)

Chi square test: ^a^,^b^ = *p <* 0.05.

**Table 7 cells-11-03611-t007:** Effects of mIVM in alginate-filled supports on MII sheep oocyte mitochondria distribution pattern.

Alginate	n. of MII Analyzed Oocytes (Runs)	Mitochondria Distribution Pattern n. (%)	
		Finely Granular	Perinuclear/Subcortical
CTRL	37 (3)	17	20
		(46)	(54)
+	34 (3)	22	12
		(65)	(35)

Chi square test: Not significant.

## Data Availability

Data are only reported in this paper.

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
