# Peer review of "Investigating and Modelling an Engineered Millifluidic In Vitro Oocyte Maturation System Reproducing the Physiological Ovary Environment in the Sheep Model"

_cells, 2022, doi:10.3390/cells11223611_

Round 1
Reviewer 1 Report
General comments
The Authors evaluated the effect of dynamic in vitro systems for maturation of lamb oocytes. Nuclear chromatin configuration, mitochondria distribution and activity, and reactive oxygen species (ROS) localization and levels were evaluated. The Authors concluded that millifluidic IVM in biocompatible inserts preserves oocyte maturation and mitochondria pattern and improves bioenergetic/oxidative status ensuring a more physiological oxygen diffusion.
The manuscript is clearly written and fully understandable, the experimental design is well explained, and the conclusions are supported by the results. However, some sentences in the discussion are too speculative (see specific comments). The manuscript is within the scope of the journal, and it represents a contribution to the current literature in this field.
In my opinion the manuscript is acceptable after very few minor revisions.
Specific comments
I would suggest adding the name of the species (sheep) in the title and in all the figures and table legends.
INTRODUCTION
Lines 70-71: the following sentence “To the best of our knowledge, the use of microfluidic culture during IVM was tested to date only on immature oocytes retrieved from female mice subjected to induced superovulation” should be edited taking into account the following paper in which immature bovine oocytes were matured in a microfluidic device:
Berenguel-Alonso M, Sabés-Alsina M, Morató R, Ymbern O, Rodríguez-Vázquez L, Talló-Parra O, Alonso-Chamarro J, Puyol M, López-Béjar M.
Rapid Prototyping of a Cyclic Olefin Copolymer Microfluidic Device for Automated Oocyte Culturing. SLAS Technol. 2017 Oct;22(5):507-517. doi: 10.1177/2472555216684625.
This reference should be also reported in the discussion.
In alternative the sentence could be rephrased as follows: “To the best of our knowledge, the use of dynamic microfluidic culture during IVM was tested…” In this case there is no need to add the above cited reference.
MATERIALS AND METHODS
Lines 188-189: “The percentage of expanded COCs, as a biomarker of oocyte maturation, was noted”. I would not consider a biomarker of maturation the expansion of the cumulus, but only a response of immature oocytes to the presence of gonadotrophins in the medium which does not ensure that the maturation is achieved.
RESULTS
Line 346: change “indreased” with “increased”.
Figure 2: histogram of ROS level is not as colorful as the others.
DISCUSSION
Lines 478-480: This is an example of speculative sentence “Moreover, the ovary native architecture has to be considered by replicating 3D curved conditions. Indeed, concave support filled with alginate gel, successfully recapitulated a portion of the ovary.” In my opinion this conclusion is an overstatement.
Author Response
General comments
The Authors evaluated the effect of dynamic in vitro systems for maturation of lamb oocytes. Nuclear chromatin configuration, mitochondria distribution and activity, and reactive oxygen species (ROS) localization and levels were evaluated. The Authors concluded that millifluidic IVM in biocompatible inserts preserves oocyte maturation and mitochondria pattern and improves bioenergetic/oxidative status ensuring a more physiological oxygen diffusion.
The manuscript is clearly written and fully understandable, the experimental design is well explained, and the conclusions are supported by the results. However, some sentences in the discussion are too speculative (see specific comments). The manuscript is within the scope of the journal, and it represents a contribution to the current literature in this field.
In my opinion the manuscript is acceptable after very few minor revisions.
We thank the reviewer for positive comments and suggestions which gave us the opportunity to improve the quality of our manuscript. We have made changes and answered all questions, as required. Answers are provided for each point.
Specific comments
I would suggest adding the name of the species (sheep) in the title and in all the figures and table legends.
The name of the species has been added in the title and in the captions of figures and tables, as suggested.
INTRODUCTION
Lines 70-71: the following sentence “To the best of our knowledge, the use of microfluidic culture during IVM was tested to date only on immature oocytes retrieved from female mice subjected to induced superovulation” should be edited taking into account the following paper in which immature bovine oocytes were matured in a microfluidic device:
Berenguel-Alonso M, Sabés-Alsina M, Morató R, Ymbern O, Rodríguez-Vázquez L, Talló-Parra O, Alonso-Chamarro J, Puyol M, López-Béjar M.
Rapid Prototyping of a Cyclic Olefin Copolymer Microfluidic Device for Automated Oocyte Culturing. SLAS Technol. 2017 Oct;22(5):507-517. doi: 10.1177/2472555216684625.
This reference should be also reported in the discussion.
In alternative the sentence could be rephrased as follows: “To the best of our knowledge, the use of dynamic microfluidic culture during IVM was tested…” In this case there is no need to add the above cited reference.
We thank the review for the suggestion. We modified the text and inserted the suggested reference.
MATERIALS AND METHODS
Lines 188-189: “The percentage of expanded COCs, as a biomarker of oocyte maturation, was noted”. I would not consider a biomarker of maturation the expansion of the cumulus, but only a response of immature oocytes to the presence of gonadotrophins in the medium which does not ensure that the maturation is achieved.
The sentence has been modified, as suggested.
RESULTS
Line 346: change “indreased” with “increased”.
The word “indreased” has been replaced with “increased”.
Figure 2: histogram of ROS level is not as colorful as the others.
We apologize for the inconvenience, but we uploaded a colour image. In any case, in the revised version of the manuscript, we substituted the previous image with a new one and hope now it works.
DISCUSSION
Lines 478-480: This is an example of speculative sentence “Moreover, the ovary native architecture has to be considered by replicating 3D curved conditions. Indeed, concave support filled with alginate gel, successfully recapitulated a portion of the ovary.” In my opinion this conclusion is an overstatement.
We thank the reviewer for the comment. The sentences have been revised or partly deleted.
Reviewer 2 Report
The revived manuscript describes application of millifluidic system for in vitro maturation of sheep oocytes. The experiment involved systems with 3D printed supports (inserts) imitating those of the ovarian follicle. The authors conclude that the applied system may improve the quality of sheep oocytes maturing in vitro by protecting them from a direct impact of the medium flow and elevated oxygen level, and also by more optimized nutrition (nutrient supply and metabolite removal). The presented data are considered preliminary and will be expanded.
I find the MS very interesting to the embryologists since a dynamic system for oocyte maturation and embryo culture has been a subject of investigation for several years. Besides, the efficiency of the static (standard) system for oocyte/embryo culture in vitro has reached its limitation especially for the blastocyst yield (in cattle 30-35%).
Although the topic of the MS is very interesting, the way it is writing needs a careful corrections.
I recommend a major revision.
Detailed comments:
My main concerns are concentrated on 3 issues:
1) Introduction: needs to be supplemented by a summary of the current status of dynamic culture systems for oocytes/embryos and what is the original impact of this experiment. Describe the new aspects of the systems you applied
2) M&M- this section needs a basic ordering. There are 3 experiments and several milifluidic systems e.g. multifluidic IVM = mIVM; 3D-mIVM in alginate layer, IVM in alginate-filled supports, mIVM with 3D printed inserts like: ring, concave + ring, Concave+ring+alginate, box; native dynamic IVM, native mIVM, native COCs, IVM by using the ring-shaped support; IVM in the presence of the open biocompatible supports concave + ring. In my opinion the single milifluidic system is described in various ways what causes a big confusion. For this reason I suggest to implement an unequivocal, clear system of nomenclature explained at the beginning of the M&M section
3) Discussion: should be focused on the novel findings of this study and clearly present the advantages/disadvantages of the applied milifluidic systems.
Author Response
The revived manuscript describes application of millifluidic system for in vitro maturation of sheep oocytes. The experiment involved systems with 3D printed supports (inserts) imitating those of the ovarian follicle. The authors conclude that the applied system may improve the quality of sheep oocytes maturing in vitro by protecting them from a direct impact of the medium flow and elevated oxygen level, and also by more optimized nutrition (nutrient supply and metabolite removal). The presented data are considered preliminary and will be expanded.
I find the MS very interesting to the embryologists since a dynamic system for oocyte maturation and embryo culture has been a subject of investigation for several years. Besides, the efficiency of the static (standard) system for oocyte/embryo culture in vitro has reached its limitation especially for the blastocyst yield (in cattle 30-35%).
Although the topic of the MS is very interesting, the way it is writing needs a careful corrections.
I recommend a major revision.
We thank the reviewer for comments and suggestions which gave us the opportunity to improve the quality of our manuscript. We have made the required changes.
Detailed comments:
My main concerns are concentrated on 3 issues:
- Introduction: needs to be supplemented by a summary of the current status of dynamic culture systems for oocytes/embryos and what is the original impact of this experiment. Describe the new aspects of the systems you applied
The Introduction has been supplemented by a summary on the current status of dynamic culture systems for most of cells and tissues of female reproductive tract, particularly oocytes/embryos. Please, find this in lines 91-102 with references 34-49. The introduction has been also revised to better clarify the new aspects of the used IVM system. Essentially, it deals on the use of a millifluidic devise which was adapted to COCs of large animals, placed and cultured in different conditions (presence/absence of biocompatible supports and gel matrix) and examined for nuclear and cytoplasmic characteristics of their developmental competence. This kind of culture will contribute to improve IVM and IVP, particularly for applications on oocytes available in suboptimal conditions, such as those aged or cryopreserved.
- M&M- this section needs a basic ordering. There are 3 experiments and several milifluidic systems e.g. multifluidic IVM = mIVM; 3D-mIVM in alginate layer, IVM in alginate-filled supports, mIVM with 3D printed inserts like: ring, concave + ring, Concave+ring+alginate, box; native dynamic IVM, native mIVM, native COCs, IVM by using the ring-shaped support; IVM in the presence of the open biocompatible supports concave + ring. In my opinion the single milifluidic system is described in various ways what causes a big confusion. For this reason I suggest to implement an unequivocal, clear system of nomenclature explained at the beginning of the M&M section
We carefully checked the nomenclature, and we made it uniform. Moreover, we inserted in M&M section a table in which the used nomenclature has been described, as it follows:
|
Name |
Description |
|
Control |
Static IVM in 4-well dish with native COCs with no artificial coatings. |
|
Native |
Dynamic mIVM in bioreactor with native COCs with no artificial coatings. |
|
Ring |
Dynamic mIVM in bioreactor with COCs inserted ring support placed at the bottom of the bioreactor chamber. |
|
Concave + ring |
Dynamic mIVM in bioreactor with COCs inserted in concave support placed at the bottom of the bioreactor chamber, with the ring support placed on its top. |
|
Concave + ring + plain = Box |
Dynamic mIVM in bioreactor with COCs inserted in a box obtained by sequentially assembling the concave support at the bottom of the bioreactor chamber, the ring on its top and the plain support used as a lid. |
|
Concave + ring + alginate = Alginate-filled supports |
Dynamic mIVM in bioreactor with COCs embedded in alginate gel loaded in concave support and ring inserted at the bottom of the bioreactor chamber. |
3) Discussion: should be focused on the novel findings of this study and clearly present the advantages/disadvantages of the applied milifluidic systems.
The advantages and disadvantages of the applied millifluidic system have been presented in the discussion. In detail, to summarize, the best analyzed condition was the concave + ring + alginate configuration (namely, alginate-filled support), as oocytes matured in this condition showed improved bioenergetic/oxidative status compared with controls.
Reviewer 3 Report
The authors investigated an in vitro millifluidic oocyte maturation (mIVM) system, in a transparent bioreactor integrated with 3D printed supports. Cumulus-oocyte complexes (COCs) from slaughtered lambs were cultured with different conditions. Nuclear chromatin configuration and mitochondria distribution pattern and activity of in vitro matured oocytes were assessed. They concluded that mIVM with COCs from large animal models takes advantage from the presence of biocompatible supports protecting
the COCs from direct medium flow and elevated oxygen concentrations.
In consideration of COCs from lambs, I agree with accept.
This is a new interesting method for oocyte IVM. More assays need to be performed to examine the oocyte quality, such as apoptosis, early embryo development.
Author Response
The authors investigated an in vitro millifluidic oocyte maturation (mIVM) system, in a transparent bioreactor integrated with 3D printed supports. Cumulus-oocyte complexes (COCs) from slaughtered lambs were cultured with different conditions. Nuclear chromatin configuration and mitochondria distribution pattern and activity of in vitro matured oocytes were assessed. They concluded that mIVM with COCs from large animal models takes advantage from the presence of biocompatible supports protecting
the COCs from direct medium flow and elevated oxygen concentrations.
In consideration of COCs from lambs, I agree with accept.
This is a new interesting method for oocyte IVM. More assays need to be performed to examine the oocyte quality, such as apoptosis, early embryo development.
We agree that other assays will be interesting to assess oocyte quality. In the present study, we focused on oocyte nuclear and cytoplasmic maturation and effects on embryo development will be the subject of further investigations.